# Management of Wild Edible Fungi in the Meseta Purépecha Region, Michoacán, México

**Eva Itzel Castro-Sánchez** [1], **Ana Isabel Moreno-Calles** [2,*] , **Sue Meneses-Eternod** [2],
**Berenice Farfán-Heredia** [3], **José Blancas** [4] **and Alejandro Casas** [5]

1   Facultad de Biología, Universidad Michoacana de San Nicolas de Hidalgo, General Francisco J. Múgica S/N A-1, Felicitas de Rio, Morelia 8030, Michoacán, Mexico

2   Escuela Nacional de Estudios Superiores Unidad Morelia, Universidad Nacional Autónoma de México, Antigua Carretera a Pátzcuaro #8701, Morelia 58190, Michoacán, Mexico

3   Centro de Cooperación Regional para la Edución de Adultos en America Latina, Área Académica Desarrollo Sustentable, Avenida Lázaro Cárdenas # 525, Colonia Revolución, Pátzcuaro 61609, Michoacán, Mexico

4   Centro de Investigación en Biodiversidad y Conservación, Universidad Autónoma del Estado de Morelos, Avenida Universidad #1001, Colonia Chamilpa, Cuernavaca 62209, Morelos, Mexico

5   Instituto de Investigaciones en Ecosistemas y Sustentabilidad, Universidad Nacional Autónoma de México, Antigua Carretera a Pátzcuaro #8701, Morelia 58190, Michoacán, Mexico

*   Correspondence: isabel_moreno@enesmorelia.unam.mx

**Abstract:** Ethnoecology is the study of the relationship between humans and their environments and components, including fungi. México is an exceptional setting for studying the interactions between humans and fungi, since most of the native cultures have interacted with these organisms for thousands of years. The state of Michoacán is particularly relevant, since nearly 11% of the fungi species recorded in Mexico occur there, 139 of which are edible. The aim of this study was to analyze the interactions of the Purépecha native communities with edible fungi and their environment, the position of mushrooms in the local worldview, and the classification system, management forms, and social and environmental problems associated with their use. Semi-structured interviews at regional markets were conducted. We conducted participant observation, proof interviews in harvesting areas, and workshops with the communities. Ethnoecological information was recorded for 21 edible fungi species and the environments where people interact with these mushrooms. People called hongueros (fungi handlers) identified the following environmental problems: A loss of local knowledge, a decreasing consumption of fungi among young people, land-use changes, the illegal extraction of forest resources, deforestation, unplanned urban growth, uncontrolled fires, livestock raising, and agricultural intensification. These issues affect fungi diversity, distribution, and abundance. All these factors, in turn, affect decreasing economic incomes associated with this activity and changes in the diets of the local people. Information from this study will help local authorities and people of the community to design management strategies for maintaining the environment and fungi, strategies which aim to contribute to the sustainable use of both fungi and forests.

**Keywords:** ethnoecology; ethnomycology; San Francisco Cherán; Michoacán; Purépecha; Kosmos; Corpus; praxis complex (KCP)

## 1. Introduction

México, as one of the five mega-diverse countries of the world, is exceptionally culturally diverse, shaped by more than 12,000 years of human–environment interactions [1]. This country offers an

exceptional opportunity for the study of a vast set of interactions between local societies and ecosystems, which are the main components of biocultural diversity [2]. Poverty, food insecurity, environmental degradation, and the loss of biological and cultural diversity are some of the greatest problems México faces today [3]. Ethnoecological, ethnobiological, and ethnomycological studies may contribute to developing strategies to address food safety, the conservation of biocultural diversity, and sustainable management of natural resources and ecosystems in this country [4,5].

The traditional knowledge use and management of fungi are some of the main components of Mexican biocultural diversity [6–10]. Guzman [11,12] estimated the occurrence of about 6710 species of fungi in the country, 2800 of which being macromycetes and nearly 300 of which being edible [13]. Fungi are of great ecological, cultural, and economic importance for rural communities. Not only is a broad spectrum of mushrooms included in the diet, but fungi also represent an important source of income for many rural households. The relationship between the Mesoamerican peoples and fungi has been the subject of many ethnomycological studies since the colonial period, as documented notably by Sahagun [14] and studied by Wasson and Wasson [15], Estrada-Torres et al. [16], and Guzmán [17]. Most of these studies, however, focused on the ceremonial or ritual uses of fungi rather than on their nutritional and medicinal properties—and even less on management aspects [18].

In México, marketplaces are ideal sites for documenting the uses and preparation of fungi and other non-timber forest products, their cultural and economic importance, and local knowledge about these resources [19–21]. In collecting fungi for consumption and for trade, many households and communities use and reproduce centuries-old knowledge and practices. Despite the ongoing and relentless acculturation of indigenous communities throughout México, trade at marketplaces contributes to the continuance of management practices [20,21]. However, not all Mexican indigenous groups consume fungi. Several studies have identified that, although most Mesoamerican cultures are mycophylic, others are mycophobic—they consider fungi harmful [22].

Mycological studies in the state of Michoacán have recorded 74 families, 652 species, and 18 infra-specific taxa of fungi [23]. Ethnomycological studies have been conducted mainly at the Monarch Butterfly Biosphere Reserve, the Patzcuaro Lake basin, the Tancítaro volcano and neighboring mountains, Tacámbaro, and the vicinity of Morelia [7,24–29]. These studies documented the classification systems, ecology, and use of fungi by local people [30]. From the species recorded, 139 are edible, 38 are poisonous, seven have medicinal uses, 152 establish symbiotic associations with other forest species (mycorrhizae), and 13 have hallucinogenic properties [31].

Though at least 43 communities have climatic and biophysical conditions suitable for the development of fungi in the Meseta Purépecha (Purépecha Plateau) region, ethnomycological studies remain scarce. Some research has been completed in the communities of Arantepacua, Sevina, and Carapan [30–32], where the traditional management practices of NTFP (non-timber forest products), particularly fungi, have been severely affected by cultural change processes. Such processes have been accompanied by the illegal extraction of wood and the over-exploitation of forest resources, as well as land-use changes associated with the expansion of agriculture, livestock, and urban settlements. Another major problem determining general biodiversity decline is climate change [33]. Given this situation, research on local environmental knowledge, the place of fungi in peoples' worldviews, and the management techniques they have developed to preserve fungi and other ecosystems is pertinent to designing new strategies for sustainable management.

In the community of San Francisco Cherán in the Purépecha Plateau region, where the dominant vegetation are pine and oak forests, fungi are abundant, but mycological and ethnomycological studies have yet to be conducted. Nearly 97% of people are Purépecha, but only 25% of them speak the native language [34]. We sought to document the existing mycological richness and its associated traditional knowledge, including the use, management techniques, main problems people perceive in the conservation of forests resources, and the solutions they and their academic allies have identified. We conducted our research within the framework of ethnoecology and strived to contribute

to the sustainable use and conservation of the community's ethnomycological patrimony through the production of useful information for local decision-making.

## 2. Materials and Methods

### 2.1. Study Area

The Purépecha Plateau is in the central-western region of the state of Michoacán in the Trans-Mexican Volcanic Belt, México. It is home to 43 indigenous communities distributed across 11 municipalities. Cherán, Carapan, Nahuatzen, and Paracho are those with the highest concentrations of indigenous population [34]. In a transitional area between the Pátzcuaro Lake and the Purépecha Plateau regions is the Paracho market where, from May to September, people go every Sunday to exchange the mushrooms they have collected for consumption.

The community of San Francisco Cherán is in the municipality of Cherán in the state of Michoacán, México (Figures 1 and 2). Cherán extends over an area of 221.88 km$^2$ at elevations between 2200 and 3200 m.a.s.l. The climate is temperate, with the annual rainfall averaging 1100 mm. Nearly 32.9% of its territory is dedicated to agriculture, with forests representing 62.01%, grasslands representing 2.38%, and the urban area representing 2.45% [35]. Forests are predominantly pine, oak, and a small portion of oyamel fir (*Abies religiosa*). Some subtropical scrub remains, dominated by huizaches (*Acacia* spp.) and mezquite (*Prosopis* spp.) trees.

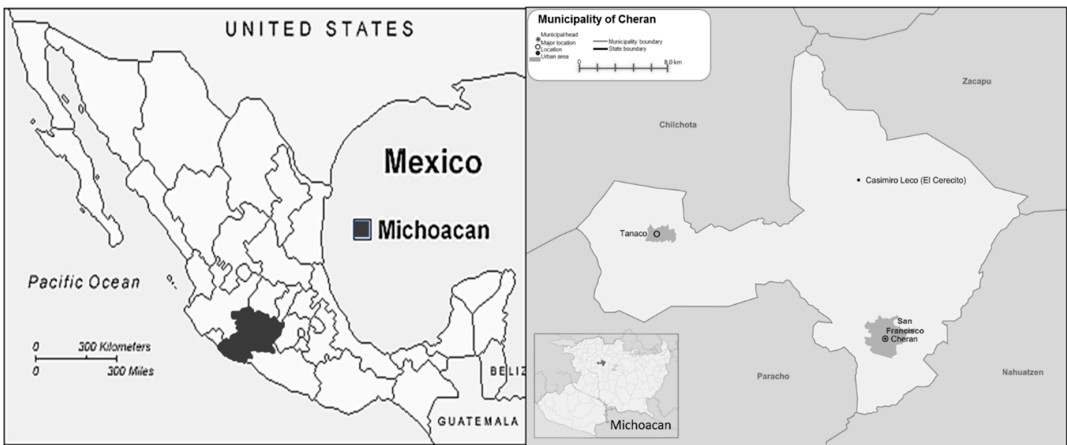

**Figure 1.** Location of the community of San Francisco Cherán, in the state of Michoacán, México.

Among the existing studies on edible wild mushroom species in the Purépecha region, those by Mapes and Caballero [6], Larios-Trujillo et al. [21], Díaz-Barriga [25], Carlos-Santos [30],Torres-Gómez [31], García-Chávez and Chávez Ramírez [32], León-Jaimes [35], Zamora-Equihua [36], Gómez-Reyes [37], and Farfán-Heredia et al. [38] are among the most relevant. These studies documented the richness of fungi in the area and called for continuing further extensive ethnomycological studies (Table 1).

**Table 1.** Ethnomycological studies carried out in the Meseta Purépecha, Michoacán.

| Reference Number | Title | Description |
|---|---|---|
| [36] | Inventory of wild edible mushrooms from the community of El Aguacate south of the municipality of Tancitaro, Michoacán, Mexico. | Exploration of local mycological knowledge as well as the relationship between *Russula brevipes* and *H. lactifluorum*. The study found 16 species of edible wild fungi and described the common names of the parts of the fungi, as well as the related worldview. |
| [39] | Knowledge and popular use of wild macromycetes in the community of Arantepacua, a municipality of Nahuatzen, Michoacán, Mexico. | Compilation of 27 Purépecha names (i.e., 16 wild fungi species and 11 of the parts of an agarical) and of the most popular species, indicating the amount of extraction, their culinary value, and their preparation methods. |
| [30] | Ethnomycology and ecological aspects of edible mushrooms from San Juan Charapan, Michoacán. | Local ecological knowledge of edible wild fungi. An estimate of their diversity, abundance, and productivity in forest areas, as well as the influence of land-use changes and deforestation on the decline of fungi productivity. |
| [32] | Didactic guide for the identification of wild mushrooms from the community of Sevina, Michoacán. | Local ecological knowledge of edible wild fungi. An estimate of their diversity. Fungi distribution in nine types of vegetation: Conserved pine, disturbed pine, conserved oak–pine, oak–juvenile pine, tejocotal (*Crataegus mexicana*) scrub, adult cedar, juvenile pine, adult pine, trunks, milpa (maize, beans, and pumpkin polyculture), and slope. |
| [38] | Ethnoecology of the interchange of wild and weedy plants and mushrooms in Phurépecha markets of Mexico: Economic motives of biotic resources management. | Analyzed the relationships between the amounts of products exchanged, which were considered pressures on the resources; the perception of their abundance or scarcity, considered the magnitude of risk in relation to the pressures referred to; and the management types as a response to pressures and risk. |
| [21] | Local knowledge and economical significance of commercialized wild edible mushrooms in the markets of Uruapan, Michoacán, Mexico. | The most culturally significant species that are sold in markets of Uruapan, Michoacán, Mexico and the reasons behind these given values using frequency and order of the mentioned indexes. |

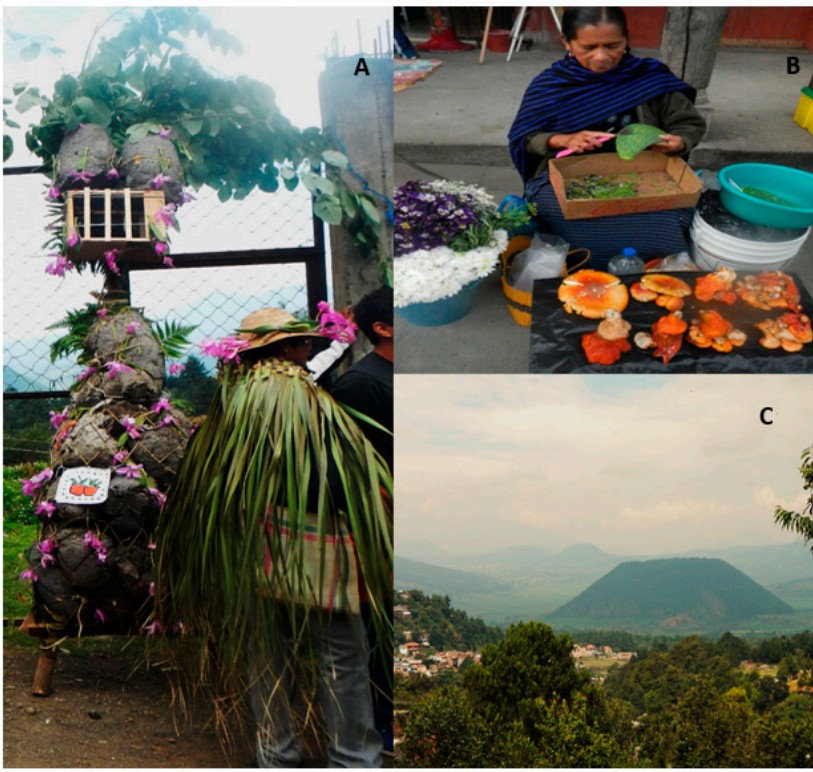

**Figure 2.** San Francisco Cherán, Michoacán, México. (**A**) The dress of men for the Corpus Christi festivities. The wild diversity of the forest or monte is shown in structures called karatakuas (at the left side of man). The fungus of iarin (*Neolentinus* spp.) was identified in these structures as part of the non-cultivated diversity that is related to humans. (**B**) A Purépecha woman selling forest flowers and mushrooms (complex of *Amanita casearea* and *Hypomices lactifluorum*) in the main square of Cherán. (**C**) Overview of the community of San Francisco Cherán.

## 2.2. Qualitative Analysis

Our research was conducted in two stages. To identify the communities that participated in the commercialization of wild mushrooms in the region and to obtain details about the exchange, information on the management and use of these forest products from May to September 2014 (the fungi harvest season) was collected. We visited the Paracho market weekly during weekends to interview sellers and buyers of mushrooms. We conducted 18 proof interviews at the market to document the number of sellers of mushrooms from each community, the richness of edible wild mushrooms sold, and the vegetation types from which mushrooms were collected. Our results identified San Francisco Cherán as a representative community for deeper further studies. In July 2014, we moved to the Cherán market and community, where we continued our studies until September of that year. We then resumed our study in 2015, from May to September. Prior to conducting interviews, a questionnaire was designed, including topics about mushrooms types, preferences, amounts, frequencies, prices, and seasonality, among other issues.

Before embarking on our research in San Francisco Cherán, we explained the project and requested a permit to perform it from the local Concejo de Bienes Comunales (Council of Communal Property), a traditional form of government that is currently still active. Notably, as agreed, the coordinates of the sites sampled are not specified, as requested by the council.

Interviewees were selected using the snowball method, through which an informant or group of key informants led to other individuals with relevant information [40]. Voice recordings were collected with the authorization of the interviewees. We conducted 18 proof interviews and guided tours to mushroom collection places with specialists (hongueros), with whom we later developed a workshop with photographs of different mushroom species for brainstorming on how to face resource shortages. The workshop focused on questions about the extraction techniques and management of edible fungi, as well as people's perception of the ecological and economic importance of this resource and its place in the agricultural and ritual calendar [41]. Information was analyzed using the Atlas ti 8 software [42]

## 2.3. Collection, Preservation, and Identification of Fungi Specimens

Mushroom specimens obtained in the markets of Paracho and Cherán, as well as those collected in the field, were photographed and used as support for further identification. For mushroom collection and preliminary identification, we followed the techniques by Cifuentes et al. [43] and Frutis-Molina and Huidobro-Salas [44]. For each specimen, we tagged aspects relevant to their identification. Species identification was first conducted at the macroscopic level, based on the taxonomic keys reported by Guzmán [45], to corroborate their identity and published illustrated guides [28,46–48]. MycoBank [48] was used to verify the correct scientific nomenclature of species. Subsequently, we followed the appropriate preservation methods [46–49]. For the correct spelling of the Purépecha names, we received the valuable support of a specialist in the Purépecha language (Sue Menesses).

## 2.4. Community Feedback

In 2016, the results of our research were shown to the community in both writing, (a thesis) provided to the Council of Communal Property, and orally, at the Second Colloquium "Juje Cheráni amperi uantaxeni" ("Let's talk about Cherán"), which was organized by the members of the community for the presentation of results from all research projects conducted in the area by different academic entities in México that year. These activities allowed feedback from local people and researchers on the information documented.

## 3. Results and Discussion

### 3.1. Paracho Municipal Market

In 2014 and 2015, sellers of edible wild mushrooms (mostly women) regularly arrived at the Paracho market from the communities of Cheránástico, San Francisco Pichátaro, San Benito Palermo,

and San Francisco Cherán. That year, there were 44 mushroom sellers (56% of the total) who did not have established positions in the Paracho and the San Francisco Cherán markets (Figure 3). They were chosen for the study.

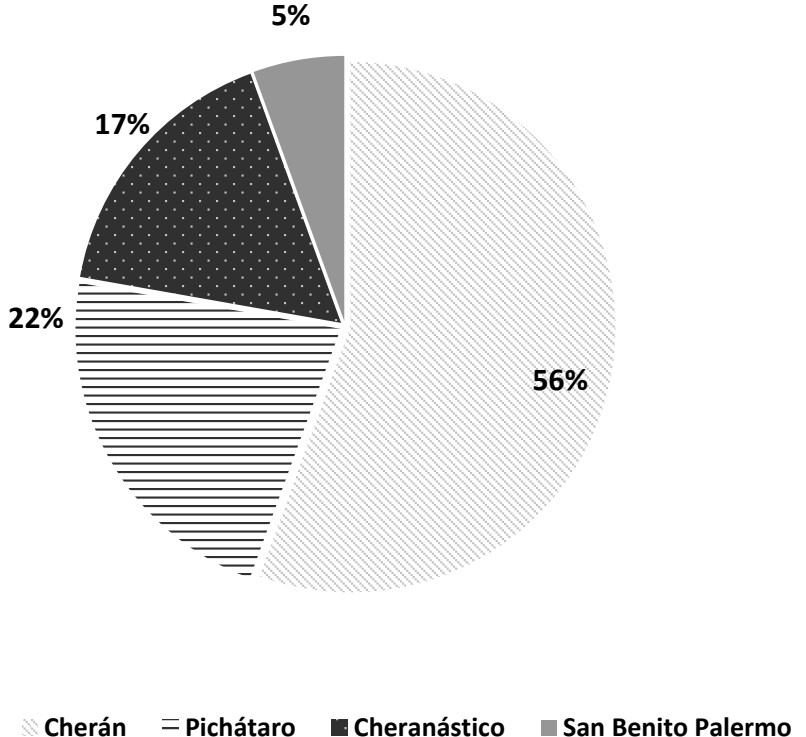

**Figure 3.** Percentage of sellers of wild edible mushrooms from the communities present in the regional market of Paracho de Verduzco during the 2014 and 2015 rainy seasons.

The main species traded were *Lyophyllum descastes*, *Lyphyllum loricatum* (pashakuas, uáchikuas), *Helvella crispa*, *Helvella lacunosa* ("mouse ears"), *H. lactifluorum* ("pork trunk"), *Ramaria flava*, and *Ramaria rubiginosa* ("bird's feet"). Because the selling of mushrooms and home garden products (e.g., medicinal plants, fruits, and other traditional greens and vegetables) is mainly a women's activity, most of the interviewees were women between 47 and 57 years old. The interviewees commented that the participation of young people in markets has visibly decreased, probably due to migration abroad.

Our interviewees explained that women have taken on the mushroom trade because it provides them the opportunity to sell garden products, handcrafts, preserves, and bread. Household interviews revealed that people who collect mushrooms may need up to three hours to travel to the collection areas, whereas the harvesting may last up to nine hours; therefore, this activity may take 12 h in total every third day (i.e., the time considered by people to let the immature mushrooms reach the adequate stage to be harvested, as an expert honguero never cuts all mushrooms but only the ripe ones. As with trade, mushroom gathering has less people and fewer young individuals participating in this activity than before. According to our interviewees, the labor force for gathering mushrooms has decreased because of migration and uncontrolled fires in forest ecosystems, which explains the decline of the mushroom trade.

### 3.2. Ethnoecology of Mushrooms in Cherán, Michoacán

#### 3.2.1. Corpus: Environmental Knowledge

Terekua, Jongo, and Jeramba

In Cherán, mushrooms in general are referred to as jongo (derived from the Spanish term hongo for mushroom) or terekua (plural terekuicha), and non-edible fungi are called jeramba. In the communities of the Pátzcuaro Lake shoreline, people place mushrooms different to the plants and animals [6]. Gómez-Reyes [32] mentioned that several communities of the Purépecha Plateau assign the generic name terekua to mushrooms, which refers to the soil or the tree root formation. In Cherán, toxic mushrooms are called jeramba terekua or, simply, jeramba. The word has the same root as jerachi or erachi, which means "brother." Adding the suffix -emba indicates the possession of the third person, so both these terms mean "their brother" [50]. This term means that mushrooms are "bad" (i.e., not useful, or poisonous) [6]. In another Purépecha dictionary, the term jeramba means "wild" [51]. In the community of Sevina, people do not use the term jeramba [37].

Purépecha Nomenclature of Edible Fungi, Species Richness, and Preferences

We identified 21 fungi species, 15 of them edible and six of them poisonous according to the locals (Table 2). This figure might seem a relatively low number of species when compared to other studies, such as Mapes et al. [6], who registered 43 edible species of fungi and 53 Purépecha names in the Patzcuaro Lake region. However, when compared to other communities of the Purépecha Plateau, the number of species recorded is relatively high. For instance, in Arantepacua, 16 species were found, and 14 were found in Carapan [30,31]. Such differences may be due to the greater extent of forests in the territory of Cherán and the efforts of civil society to defend and protect the forest. Table 2 provides scientific and Purépecha local names for the edible and poisonous fungi species. Additional information [6,52], as well as from interviews of people in the Pichátaro and San Francisco Cherán markets, was included in this study.

According to most of the interviewees in the community of San Francisco Cherán (96%), there are six poisonous species of mushrooms, although three of them (*G. dryophilus*, *H. lacunosa,* and *R. brevipes*) are considered edible in other communities of the Purépecha Plateau, namely Arantepacua, Sevina, and Carapan [30,31,37]. *Cantharellus cibarius* is controversial. Whereas people in Cherán consider it poisonous, one family in this community and people from other communities of the Purépecha Plateau consume this species. The origin of this controversy may be that *C. cibarius* resembles *R. brevipes*, a species considered poisonous, but its color resembles that of tiámu terekua, the "pork trunk" edible mushroom (*H. lactifluorum*). In the identification guides [46–49], as well as according to local people, *A. muscaria* and *A. virosa* are reported to be poisonous.

For their abundance and good flavor, tiámu terekua (*H. lactifluorum*), tiripiti terekua (complex of *A. caesarea*), and kuinit jantsiri terekua (*Ramaria* spp.) are the most traded fungi species in Cherán. Uáshikuas (*L. loricatum*) and pashakuas (*L. decastes*), though highly abundant, are not as tasty as the three fungi mentioned above, and their price is lower (nearly USD $1 per kg). Less abundant is *H. lactifluorum*, the most expensive mushroom species in Cherán (between USD $4 and 5 per kg). Table 3 shows the preferences by flavor and income of edible wild fungi species. Some species, like *Boletus* aff *edulis*, *C. cibarius*, *C. gibba*, and *L. perlatum*, are used for home consumption, since they are scarce or little known.

**Table 2.** Scientific and Purépecha local names for the edible and poisonous fungi species in the Meseta Purépecha region, Michoacán, Mexico.

| Scientific Name | Purépecha Name/Local Name | Additional Information | Edible (E) or Poisonous (P) |
|---|---|---|---|
| *A. caesarea* (Scop.) Pers. | Tiripiti terekua, yellow | Tiripiti: "Golden;" terekua: "mushroom" | E |
| *Amanita muscaria* (L) Lam. | Chets, sweetened bread | - | P |
| *Amanita virosa* Secr. | White | - | P |
| *Boletus* aff *edulis* Bull. | Panterekua or Pan terekua, the belly, the belly of an ox, the belly of an old woman, bread | Pan terekua: Mushroom in the form of bread. From Spanish *pan* (bread) | E and medicinal |
| *Cantharellus cibarius* Fr. | Pig horn jeramba | Jeramba: "Bad," "poisonous," "wild" | E and P |
| *Clitocybe gibba* (Pers.) P.Klum | Small beans | | E |
| *Gymnopus dryophilus* (Bull.) Murrill | False small beans | - | P |
| *Neolentinus* spp. Redhead and Ginns | Yarinterekua or iarhini terekua, yarin, mushroom, little eagle | Iarini: "Mature pine," iarini terekua: Variety of edible fungus that grows on old sticks (Velasquez Gallardo, 1978) | E |
| *H. crispa* Bull. | Sirat agants or siráata angánts terekua, mouse little ears | Sïrata anha-ntsï terekua: Erect-head smoke or smoke standing on top, variety of edible fungus (now written sïrata anhantsï) | E |
| *Helvella lacunose* Afzel | Sirat agants or siráata angánts terekua jeramba | Sïrata anha-ntsï terekua jeramba: Erect-head smoke wild mushroom, "not good" | E and P |
| *H. lactifluorum* (Schwein.) Tul. and C. Tul. | Tiámu terekua, horn pig | Tiámu terekua: Tiámu "iron," terekua "mushroom" | E |
| *Laccaria laccata* (Scop.) Cooke | Small beans | - | E |
| *L. loricatum* (Fr.) Kühner | Huashikuas, pashakuas, huachikuas | Paxa-: Division of roads [51]. Uáxikuas or uáchikuas are variants of the same word and -s is the plural in Spanish, it appears as uáchitas or "small cumulus" [6]. For Diccionario Grande, uacheni means "to be many." Now, the words for "a lot" are uá-nekua and uá-ni, which have the root ua- "a lot." For Delfina Duran, inhabitant of Cherán, this means "ladies," as they are always in groups. | E |
| *Lyophyllum decastes* (Fr.) Singer | Huashikuas, pashakuas, huachikuas | | E |
| *Lycoperdon perlatum* Pers. | Caca de nana kutsi. Mother Moon poo | Nana kutsï: "Mother Moon" | E |
| *R.* aff *rubiginosa* Marr and D.E. Stuntz | Kuinit jantsiri terekua. Bird's paw coffee | Kuinitu jantsiri: "Little bird's paw" | E |
| *R. flava* (Schaeff.) Quel. | Kuinit jantsiri terekua. Bird's paw yellow (sweet smell) | - | E |
| *Ramaria formosa* (Pers.) Quel. | Kuinit jantsiri terekua jeramba. Bird's paw light yellow | - | P |
| *Ramaria* aff *flavigelatinosa* Marr and D.E. Stuntz | Kuinit jantsiri terekua. Bird's paw white | - | E |
| *R. brevipes* Peck. | Pig's trunk, jeramba | - | P |
| *Sparassis crispa* Marr and D.E. Stuntz | Oxen yoke | - | E |

**Table 3.** Preferences by flavor and incomes of edible fungi species in the Meseta Purépecha region, Michoacán, Mexico.

| Species | Preference by Flavor | Income (250 g/dollar) |
|---|:---:|:---:|
| *Amanita caesarea* | ‖‖‖ | 2.55–5.10 |
| *Boletus* aff *edulis* | ‖ | - |
| *Chantarellus cibarius* | ‖ | - |
| *Clitocybe gibba* | ‖ | - |
| *Neolentinus* spp. | ‖‖‖ | - |
| *Helvella crispa* | ‖ | 1.02 |
| *Helvella lactifluorum* | ‖‖‖ | 1.27–5.10 |
| *Laccaria loricatum* | ‖ | 1.02–2.55 |
| *Laccaria descastes* | ‖ | 1.02–2.55 |
| *Laccaria perlatum* | ‖ | - |
| *Ramaria* aff *rubiginosa* | ‖ | 1.02 |
| *Ramaria flava* | ‖ | 1.02 |
| *Ramaria botrytis* | ‖ | 1.02 |
| *Ramaria* aff *flavigelatinosa* | ‖ | 1.02 |
| *Sparassis crispa* | ‖ | 5.02–12.74 (by size) |

Symbols: ‖‖‖ very tasty, ‖ tasty, ‖ a little tasty. Prices are in USD. "-": Not available for sale or without market value.

### 3.2.2. Praxis: Management and Harvesting Techniques

In Cherán, knowledge is orally transmitted from parents to children. Grandparents and parents are key actors in the identification of edible and poisonous species, as they teach the children how to distinguish one mushroom from another. As in other communities in the Purépecha region [30,31], people identify mushrooms based on their texture, color, smell, and the place where they occur. Collection times and places are determined based on the amount of rainfall and other indicators, such as the local presence of the dual fungus jeramba, which signals the upcoming of new sprouts (according to people's perceptions).

During the rainy season, local people mention that the extraction of pine resin and the harvest of dry wood decreases, so fungi are an alternative source of food and income. People in Cherán recognize the importance of keeping trees and vegetation to ensure fungi growth, and this has helped to preserve forests in the area. They know that vegetation type and topography determine the availability of fungi and can be used to identify the different kinds of mushrooms available at the pinadas (young pines, mainly *Pinus* aff *leiophylla*), the encineras (*Quercus crassipes*), the tepamu forest (*Alnus* aff *acuminata*), the sharari forest (*Quercus* aff *laeta*), and the grasslands (Table 4).

Fungi harvesting coincides with maize cultivation between the dry and the rainy seasons (from April to October). Sometimes maize harvesting is delayed, and mouse ears mushrooms and maize are harvested until November or December. Maize is also irrigated but restricted to a relatively small area. In June, *L. decastes* and *L. loricatum* are abundant. Then, in the months of July and August, many mushroom species abound. Whereas *H. lactifluorum* and *Ramaria* spp. are predominant, there is also some complex of *A. caesarea*, as well as *L. decastes* and *L. loricatum*. Finally, *H. crispa* and *H. lacunosa* grow from the end of August to September. According to people interviewed, there may be pig trunks until November, but this species is reserved for self-consumption. In November, at the beginning of the dry season, dry wood collection and pine resin extraction become the main activities, along with the making of embroidered garments for the holidays in November, December, and January. Central to the worldview of this community are the rituals offered to the protective beings of the forest during this season (Figure 4).

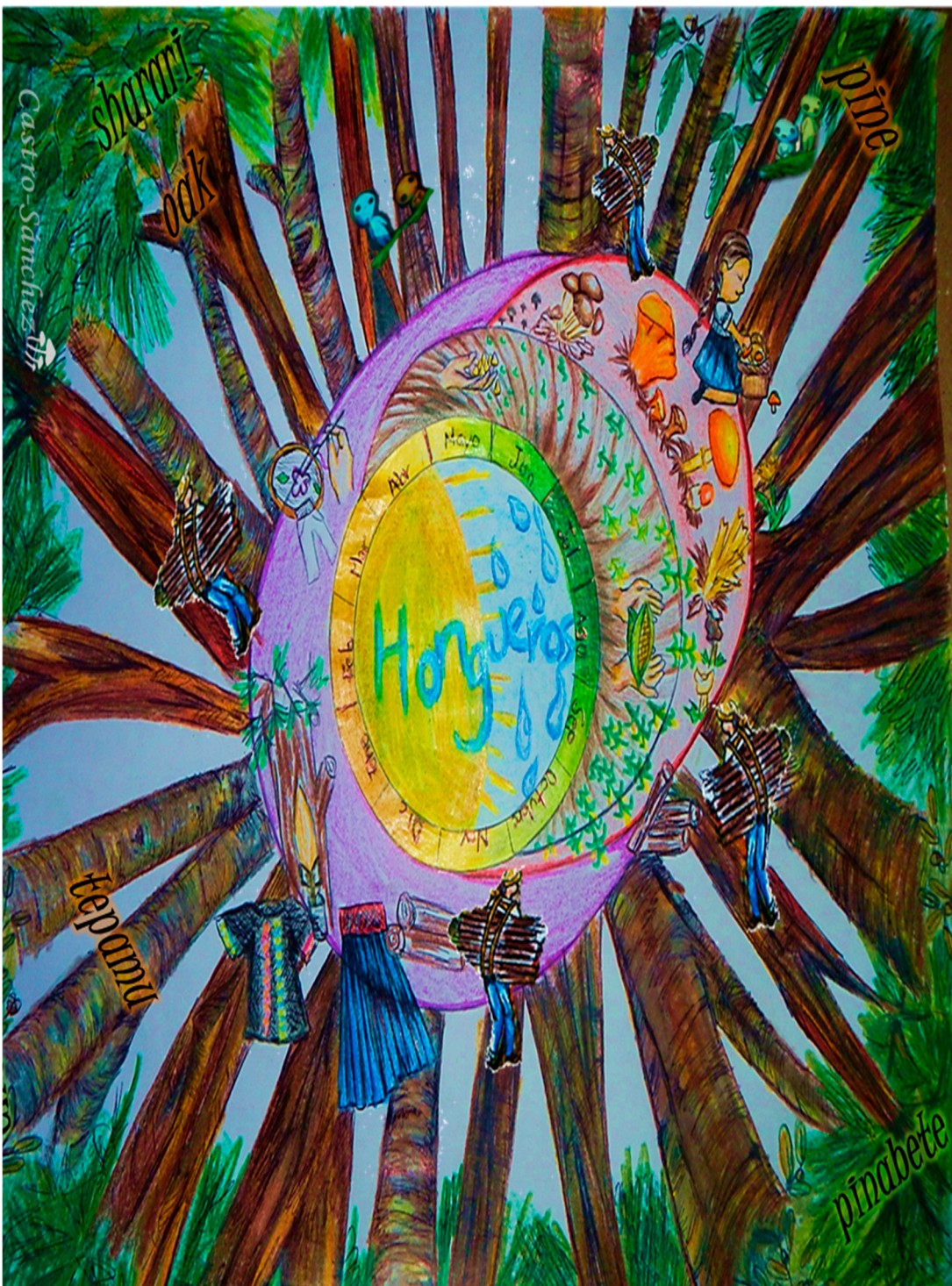

**Figure 4.** Calendar of activities related to mushroom gathering in San Francisco Cherán. The annual activities are illustrated, which are divided into two seasons: Dry and rainy. The rainy season begins with the first rains at the end of May and continues until the first weeks of September. In addition to mushroom harvesting, the cultivation of the traditional multi-crop system milpa (i.e., maize, beans, and pumpkin), the harvest of medicinal and ornamental plants, and the embroideries for the festivities of December and January are undertaken. Created by Eva Castro Sánchez.

**Table 4.** Wild fungi species and vegetation type in the Meseta Purépecha region, Michoacán, Mexico. The different vegetation types usually found in Cherán, as identified by the local people. To assess the presence and abundance of fungi, hongueros stated that they consider the vegetation type, the incidence of light, the number of tree species in the area, and the topography.

| Wild Fungi Species | Vegetation Type or Habitat |
|---|---|
| *Amanita muscaria* | |
| *Amanita virosa* | |
| *Boletus* aff *edulis* | |
| *Cantharellus cibarius* | |
| *Clitocybe gibba* | |
| *Gymnopus dryophilus* | Pinadas (Sets of young pines, mainly *P.* aff *leiophylla*) |
| *Laccaria laccata* | |
| *Ramaria* aff *rubiginosa* | |
| *Ramaria flava* | |
| *Ramaria formosa* | |
| *Ramaria* aff *flavigelatinosa* | |
| *Ramaria brevipes* | |
| *Sparassis crispa* | |
| *Neolentinus* spp. | |
| *Helvella crispa* | Rotten tree trunks Pineland/grassland/cedar forest |
| *Helvella lacunose* | |
| *Helvella lactifluorum* | Tepamu vegetation (*A.* aff *acuminata*) or Pinada |
| *Laccaria loricatum* | Holm oaks (*Q. crassipes*) |
| *Lyophillum descastes* | Sharari vegetation (*Q.* aff *laeta*) |
| *Lycoperdon perlatum* | Oaks and grassland |

Fungi Management and Harvesting Techniques

Harvesting techniques vary according to the species collected. Most collectors extract only the larger and open individuals (mature fructiferous bodies or carpophores) and give them a slight blow on the hat in order to leave the spores behind to ensure reproduction. The smaller mushrooms are left in the forest to continue growing until the next year. Due to illegal logging and land-use changes, collection areas are increasingly more distant. Figure 5 shows the different extraction techniques performed by fungi experts for the most commercialized species.

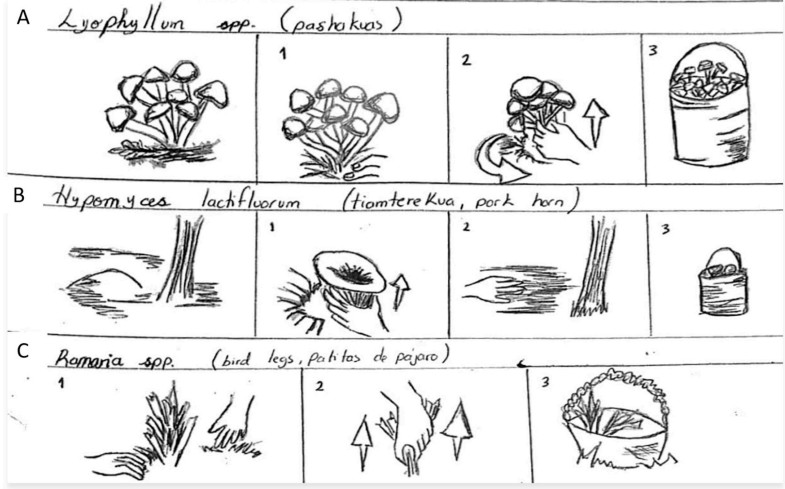

**Figure 5.** Different extraction techniques for the most commercialized wild mushrooms. (**A**) Huachikuas, pashakuas, or huashikuas (*L. descasca* and *L. loricatum*): (1) It is superficially dug with the fingers, (2) the group with more mature mushrooms (open hat) is chosen, then the whole bases of the mushrooms (stipe) rotate as they rise, and (3) they are deposited in buckets or baskets. (**B**) Horn of pig or tiamterekua (*H. lactifluorum*): These species make mounds of huinumo (mounds of dried pine leaves) and, to find the fungi, these are carefully explored. (1) To extract the fungus, it is pulled from the stipe to not break it. (2) Then, the hole is covered with pine needles so that the spores do not dry out, and (3) they are deposited in buckets. (**C**) Small legs of birds (*R. flava* and *R. rubiginosa*): (1) The area where the fungus is found is superficially cleared. (2) The fungus is removed from the base and (3) carefully placed in baskets or cans. Created by Eva Castro Sánchez.

These techniques are much like the those recorded from individuals at the Paracho market, except with regards the tiripiti terekua or yellow mushroom (belonging to the *A. caesarea* complex). They believe in cutting only the "hat" (cap), leaving the "leg" (stem) along with the "egg" (volva) in the site so it can re-sprout. Some people leave only the egg. In the case of the iarini terekua mushroom (*Neolentinus* spp.), also called, iarín or iarini terekua because it grows on the iarín (hard matter) of dead and rotten stumps of pines, a part of the stem is cut off, leaving only a couple of "fingers" on the trunk so the mushroom can re-sprout the following year. Finally, the oxen yoke mushroom is considered "shy" because, if seen, it does not grow. During the months of June to August, hongueros avoid the places where this type of mushroom grows, mostly rotten trunks in very humid areas, and they harvest them the following year. We did not find traces of this type of management and knowledge in the Purépecha Plateau. Mushroom gathering occurs from June to August, along with the collection of medicinal plants. Dry wood collection continues throughout the year, although it decreases during the rainy season. Collection in the forest is an activity that requires special permission from the Council of Communal Property, and unauthorized collectors are subject to sanctions.

### 3.2.3. Purépecha Kosmos and the Community's Relationship with the Forest

For the people of Cherán, the forest is a resource of great spiritual, social, cultural, and economic importance. They believe that the existence of forest guardian spirits, amenable beings that can either provide or deny food, depends on the relationship with the forest. La Miringua is one of these guardians, a spirit that confuses those who want to harm the forest or who, in a state of drunkenness, get lost in the forest for days; the spirit makes those who want to harm the forest get lost [53]. Due to this belief, during reforestation programs, a ritual is performed where the guardians are asked for permission to complete the reforestation. This is not only an attitude of respect and recognition of the sacred entities of the forest but also occurs with the purpose of diminishing the risk for the participants and favor the work of reconstruction of the environment [53].

### 3.3. Problems and Possible Solutions in the Management and Consumption of Fungi

The protection of natural resources in Cherán and other communities of the Purépecha Plateau is fraught with several problems. In addition to social conflicts, traditional forest activities are jeopardized by land use changes and other issues. During the interviews, among the many described problems faced by the hongueros, we identified the following main issues based on information from the local people in the Cherán community: Constant uncontrolled fires, land-use changes (i.e., avocado cultivation and urbanization), and climate change. All of these have affected the abundance of fungi they consume. In a study on land-use changes in the Purépecha Plateau in 1956, 2000, and 2005, Garibay-Orozco and Bocco-Verdinelli [54] identified avocado cultivation as the main factor responsible for this process. However, in the case of Cherán, the topography naturally limits the expansion of this crop (highlands with low temperatures), the illegal extraction of forest resources, and the advance of unplanned urbanization directly affect the forest as it similarly does. To a lesser extent, agriculture, livestock raising, and uncontrolled fires affect the forest. All these factors contribute to massive amounts of deforestation and soil erosion, as well as to the reduction of native vegetation and rainwater retention, which both reduce the diversity and the abundance of wild fungi.

In the case of fires, the hongueros have seen that the abundance and richness of the already known fungi have been changed and have favored an increase in other fungi that they do not know. This is because fire is perceived as a factor affecting the survival of mushroom "seeds" (spores), which may favor the development of non-edible species. Fires favor the growth of weeds to the detriment of some fungus and pine species. The damage varies with the intensity of the fire and the species concerned. Most affected are saprophytic fungi located at the surface of the soil, as well as the mycorrhizal communities that grow on the radial tissues of superficial tree roots. The intensity of a fire depends on various factors, such as soil moisture, the amount of dry organic matter and its caloric capacity, the topography, the temperature, and the speed of wind; however, some fungus

species may benefit from fire [55]. For instance, Gómez-Reyes et al. [56] found that fires can activate germination and promote the development in burned sites of exclusive species in a state of latency in Barranca de Cupatitzio, Uruapan, and, depending on their frequency and intensity, fires may alter the plant cover of temperate forests. Harvesting and selling fungi now require more time, dedication, and effort. The task of identifying toxic species from edible species requires specialized people, the numbers of whom are in sharp decline due to the preference of other activities by young people, including work outside the municipality and even abroad [36]. Another major problem identified by people in this community is the erratic behavior of the rain, something attributed to the decline in forest density and global climate change. Apart from reforestation, other solutions have been proposed, such as inviting young people with enough knowledge and interest in fungi to participate in different activities, where they can share their experiences with other youngsters. This may be supported by information talks about the importance of the activity and the nutritional properties of fungi and its ecological importance in ecosystems. These talks may include the participation of authorities, community members, and academic experts. It is also possible to experiment with the cultivation techniques for some species, as is being done by some local people and scholars. Finally, gastronomic sampling and myco-tourism have also been proposed. All these elements are important to the construction and planning of a communitarian strategy for the preservation of forests and to ensure the availability of mushrooms, even increasing their use, while preserving the knowledge and tradition around fungi in the Purépecha Plateau.

## 4. Conclusions

The information derived from this study will help to collectors, authorities, and scientists to design strategies for the conservation of people, the environment, and fungi diversity. There is a wealth of edible wild mushrooms in the Purépecha Plateau and in San Francisco Cherán compared to other localities with similar ecological–cultural conditions. The inhabitants of this region use the mushrooms for medicine, food, income, and ritual purposes. The knowledge of hongueros allows them to distinguish edible mushrooms from toxic ones, but this knowledge needs to be spread to the community, particularly young people.

Mushroom management has so far ensured the continuity of edible fungi, although socio–environmental factors, such as migration and changes in food habits and cultural and economic patterns, affect the conservation of native vegetation and fungi and the ability of people to access them. The interactions between human beings and fungi are complex, and these interactions are the focus of ethnoecology. The problems of the relationships of human beings, particularly of native peoples, with fungi and their environments are problems of sustainability. These problems need to be understood through the different views of these groups to design inclusive and relevant alternatives in these contexts.

**Author Contributions:** E.C. is the main author, and A.I.M.C. is the thesis research project director. They contributed to the development of this project, from its inception to the writing of this report, through planning, fieldwork, interviews, and data analysis. S.M. is a Purépecha language professor. B.F., J.B. and A.C. drafted and reviewed the manuscript.

**Funding:** The authors acknowledge the support of the DGAPA-UNAM PAPIIT IN200417 and the PAPIME PE209517 projects in the study's design; the collection, analysis, and interpretation of data; and the writing of this manuscript.

**Acknowledgments:** The authors thank the Concejo de Bienes Comunales, Genoveva Pedroza, and the *hongueros* of San Francisco Cherán.

**Conflicts of Interest:** The authors declare that they have no competing interests.

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
