# Peer review of "Management of Wild Edible Fungi in the Meseta Purépecha Region, Michoacán, México"

_sustainability, doi:10.3390/su11143779_

Round 1

Reviewer 1 Report

The re-submitted manuscript has been significantly improved. The introduction and result and discussion are much scientific sound. It is acceptable for publishing after some copyediting. 

a couple minor concerns:

1. In 3.1 section first paragraph, it described the situation in 2014 for figure 3, but fig3 legend indicated the data during 2014 and 2015. It needs to be consistent and/or to be clarified.

2. In the legend of Table 3, please indicate what the symbol ‘-‘ presents for. Is it unavailable or not for selling, or something else?

Author Response

Reviewer 1

The re-submitted manuscript has been significantly improved. The introduction and result and discussion are much scientific sound. It is acceptable for publishing after some copyediting. 

a couple minor concerns:

1.     In 3.1 section first paragraph, it described the situation in 2014 for figure 3, but fig3 legend indicated the data during 2014 and 2015. It needs to be consistent and/or to be clarified.

Authors: Done

2.     In the legend of Table 3, please indicate what the symbol ‘-‘ presents for. Is it unavailable or not for selling, or something else?

Authors: It is not sold, it is only available for self-consumption or it has no price in the market. We included this information in the legend.

Reviewer 2 Report

The quality of the manuscript became much better after revision. The authors have made a lot of improvements in the presentation of the results of their study. I recommend to accept this article for the publication in present form.

Author Response

Thank you

Reviewer 3 Report

I was pleasantly surprised to see a nice revision and resubmission of this manuscript.  It provides important documentation of ethnoecological information from a region rich in multiple forms of diversity.  The major revisions have much improved the flow and figures, and I only found minor editing proofs to comment on. 

In Section 3.1, the 2 paragraphs after Figure 3 (the second paragraph is only one sentence), should be combined.  

The editor may have a preference for this, but the words Corpus and Praxis in the start of sections 3.2.1. and 3.2.2 do not seem needed.  3.2.2 and the subheader 3.2.2.1 seem to be saying the same thing. 

It would be beneficial to write out the complete genus and species of fungi names for Tables 3, 4.  Any tables potentially taken out of context of the manuscript should be able to know what the species are.  I would also suggest adding a caveat of "in the Meseta Purepecha region, Michoacan, Mexico" to each of the table captions.

Figure 4 appears stretched, but this may be fixed by editing process. 

Figure 5 and the caption are good additions!  Please attribute the artist for 4 and 5. 

Paragraph 3.2.3.  In the sentence about La Miringua, it is unclear if it is the spirit who gets lost in the forest, or if the spirit makes those who want to harm the forest get lost.  

Section 4, first two paragraphs should be combined (first is only one sentence).    Delete the first "to" before the word collectors in the first sentence. 

Author Response

Reviewer 3

I was pleasantly surprised to see a nice revision and resubmission of this manuscript.  It provides important documentation of ethnoecological information from a region rich in multiple forms of diversity.  The major revisions have much improved the flow and figures, and I only found minor editing proofs to comment on. 

In Section 3.1, the 2 paragraphs after Figure 3 (the second paragraph is only one sentence), should be combined.  

Authors: Done

The editor may have a preference for this, but the words Corpus and Praxis in the start of sections 3.2.1. and 3.2.2 do not seem needed.  3.2.2 and the subheader 3.2.2.1 seem to be saying the same thing. 

Authors: We consider that these headings are necessary to clarify the ethnoecological components of the work, but we remain attentive to the suggestions of the editor if he does not consider them necessary.

It would be beneficial to write out the complete genus and species of fungi names for Tables 3, 4.  Any tables potentially taken out of context of the manuscript should be able to know what the species are. 

Authors: Done

I would also suggest adding a caveat of "in the Meseta Purepecha region, Michoacan, Mexico" to each of the table captions.

Authors: Done

Figure 4 appears stretched, but this may be fixed by editing process. 

Authors: Ok

Figure 5 and the caption are good additions!  Please attribute the artist for 4 and 5. 

Authors: Done

Paragraph 3.2.3.  In the sentence about La Miringua, it is unclear if it is the spirit who gets lost in the forest, or if the spirit makes those who want to harm the forest get lost.  

Authors: The answer is the spirit makes those who want to harm the forest get lost.  We chance the setence for clarify.

Section 4, first two paragraphs should be combined (first is only one sentence).    Delete the first "to" before the word collectors in the first sentence. 

Authors: Done